# Investigation of the Lubricated Tribo-System of Modified Electrospark Coatings

**Mindaugas Rukanskis** [1], **Juozas Padgurskas** [1], **Valentin Mihailov** [2], **Raimundas Rukuiža** [1], **Audrius Žunda** [1,*], **Kęstutis Baltakys** [3] and **Simona Tučkutė** [4]

1. Department of Mechanical, Energy and Biotechnology Engineering, Faculty of Engineering, Vytautas Magnus University Agriculture Academy, 53361 Kaunas, Lithuania; raimundas.rukuiza@vdu.lt (R.R.)
2. Institute of Applied Physics, Moldova State University, 2028 Chisinau, Moldova
3. Department of Silicate Technology, Faculty of Chemical Technology, Radvilėnų St. 19, 44249 Kaunas, Lithuania
4. Center for Hydrogen Energy Technologies, Lithuanian Energy Institute, Breslaujos St. 3, 44403 Kaunas, Lithuania
* Correspondence: audrius.zunda@vdu.lt; Tel.: +370-37-788149

**Abstract:** This work presents the results of tribological tests of Mo and combined coatings TiAlC formed using electrospark alloying (ESA) technology and additionally processed using thermochemical electrolyte anodic heating (TEAH). ESA makes it possible to form 15–20 μm thick coatings on the friction surface, characterized by a high hardness and wear resistance. Tribological studies were performed by testing the block-on-roll friction pair under 300 N and 600 N loads. The duration of the tests was 180 km of friction path, and the constant rotation rate of the disk was 600 rpm. It was observed that the friction torque during the 300 N loading tests was stable for all samples and throughout the entire testing period, whereas at 600 N loading, the pair with the Mo coating had a decreasing trend, and the pair with the TiAlC coating, friction torque slightly increased. For a reference sample without the coating trend of friction torque became drastically unstable. At both loads (300 N and 600 N), the friction pair with the reference sample had the highest cumulative wear, and the pair with the Mo coating had the lowest. At both loads, the cumulative wear of the friction pair with Mo coating is about 2 times lower than the TiAlC, and $\geq$1.8 times lower than the control (not coated) version. This study shows that at lower loads, the friction pair formed by the TiAlC coating and steel C45 is more matched than the friction pair with Mo coating.

**Keywords:** electrospark alloying; Mo coating; TiAlC coating; thermochemical electrolyte anodic heating; wear resistance; friction

## 1. Introduction

Technological surface coatings are an important way to strengthen the surfaces of worn machine parts. Coating technologies, such as vacuum plasma spraying, laser coating, chemical and physical vapor deposition, arc metallization, and others, are used to produce amorphous alloy coatings. Electrospark alloying (ESA) technology has been studied since the 1980s and has advantages over other coating technologies, as ESA is easy to implement with simple equipment and has significantly lower material and energy costs compared to other coating technologies. An essential advantage of ESA is excellent adhesion of the coating to the coated surface because the coating is formed by the intense mixing of the molten materials of the electrode (anode) and the base (cathode). In addition, careful surface preparation is not required before treatment, as the electric arc cleans all the dirt from the surface to be coated [1,2]. Such wear-resistant coatings on metal surfaces are formed due to many short-term electric arcs when the material of the electrode melts into tiny droplets, which are accelerated by the electric arc, and then hit the surface and alloy

with the substrate [3,4]. By controlling the selection of ESA electrical pulse parameters and their sequence, the surface roughness of the coatings can be controlled [5].

The chemical treatment during anodic electrolyte heating provides the saturation of the metal surface by light elements such as nitrogen, carbon, boron, and oxygen. Nitrohardening of the steel may be realized in the aqueous solutions of ammonia chloride. Carburizing of the steel surface is realized using acetone, glycerol, sucrose, or ethylene glycol as a component in ammonia chloride electrolytes [6].

Molybdenum plays an irreplaceable role in adjusting the hardness and toughness of steel. Different coating technologies, such as magnetron sputtering, thermal spraying, electric thermal explosion spraying, or electrospark, deposit Mo on various substrates. The obtainable coating structure is dense and uniform. In addition to this, Mo is essential to the industry due to its high melting point, high thermal conductivity, and low thermal expansion coefficient. Moreover, it is an excellent coating metal in surface modification due to its superior metallurgical bonding with various metals and alloys. As a wear-resistant coating material, Mo is extensively used in engine piston rings, bearings, seals and shafts, as an overlay coating to prevent surface damage and degradation, in papermaking, aerospace, etc. [7,8].

Recently, TiAlC-based coatings have attracted considerable attention, as they may exhibit good bearing capacity in the marine environment and ground industry in terms of good anticorrosion properties and wear resistance [9,10]. Those coatings are commonly produced by using physical vapor deposition and chemical vapor deposition. Several studies reported coatings prepared using magnetron sputtering [11]. Only a few works have researched TiAlC-based coatings formed using the electrospark method [12–14].

In this work, the samples with molybdenum and combined TiAlC coatings using electrospark deposition were nitrided and carbonized using thermochemical electrolyte anodic heating (TEAH) as a secondary treatment with the idea of enhancing the mechanical and tribological properties of the created coatings. The mechanism of anode heating is as follows. The surface area of the bath—the cathode is tens of times larger than the area of the workpiece—the anode; therefore, the main part of the source energy is released in the anode region. This process leads to the local boiling of the electrolyte and the formation of a continuous vapor-gas shell surrounding the workpiece. The intense emission of ions of a dissolved substance from a boiling electrolyte and their transfer through the paragaseous shell under the action of an electric field ensures the passage of a current of ~1 A/cm$^2$ and the release of energies of ~106 W/m$^2$, which drives the part to heat up to 500–1000 °C [15].

Based on several studies investigating its characteristics [2,13,14,16,17], we selected for our study the electrospark alloying of molybdenum and combined TiAlC coatings and their treatment with TEAH.

The present study aimed to examine the mechanical and tribological properties of different electrospark alloying coatings with TEAH processing. For this purpose, Mo and combined TiAlC ESA coatings were additionally processed using chemical treatment during anodic electrolyte heating, and the results were compared with the specimens without any coating as a reference, testing it with a hardened steel counter-body at different loads and rotation speeds in lubricated conditions.

## 2. Materials and Methods

In this study, the friction pairs used for tests consisted of three types of rectangular segments (dimensions $5 \times 10 \times 23$ mm) made from steel C45: (1) deposited by ESA molybdenum coating with TEAH treatment (Mo + TEAH), (2) combined TiAlC ESA coating with TEAH treatment (TiAlC + TEAH), and (3) reference sample (RS) made of steel C45 without any coating. Hardened steel C45 (HRC 42–45) disks (outer diameter $16^{-0.05}$, width $12 \pm 0.2$ mm) were used as a counterpart. The hardening process of the disks was as follows: heating in an oven at 830 °C for 20 min, cooling in water until reaching the ambient temperature, annealing at 400 °C for 30 min, and cooling at an ambient temperature.

Four different rods with a diameter of 5 mm and a length of 40 mm—molybdenum, titanium, aluminum, and graphite—were chosen as electrodes (Table 1). The coatings were

applied by the electrospark method using the industrial machine EFI-10M that operated in the modes close to those used in previous work depositing Mo and combined Ti-Al-C coatings, with a current intensity of 0.7–2.0 A, a pulse duration of 200–250 μs, and a voltage of 200–230 V, under the conditions of an unprotected medium [14,16]. Three layers of molybdenum, using different currents, were deposited on the segments to obtain solid and even coatings. The combined coating materials were deposited in the following order: titanium + aluminum + graphite, using different deposition currents. The electrospark deposition modes are presented in more detail in Table 1.

**Table 1.** Electrospark alloying regimes of machine EFI-10M expressed in electrospark current.

| Deposited Material | The Current of Electrospark Alloying, A |
|---|---|
| Molybdenum | 1 layer: 1.2–1.5<br>2 layer: 1.5–2.0<br>3 layer: 0.7–1.2 |
| Titanium | 1.2–1.5 |
| Aluminum | 1.5–2.0 |
| C—graphite | 1.5–2.0 |

Mo and combined TiAlC coatings were further processed by TEAH chemical treatment. The literature shows that an electrolyte containing 10% $NH_4Cl$ and 5% $NH_4OH$ ensures a relatively high nitrogen content. Using glycerol as a component of ammonia chloride, electrolyte carburizing the surface can be realized [6]. This study tried to obtain both of the benefits, so the electrolyte with the following composition was chosen: 10% of $NH_4Cl$ (ammonium chloride), 10% of $C_3H_5(OH)_3$ (glycerine), and the rest of the water. The processing time of the samples was 3 min.

The SEM pictures of Mo + TEAH and TiAlC + TEAH cross-sections and the distribution of chemical elements are presented in Figure 1.

The cross-section picture of the Mo + TEAH coating (Figure 1a) shows that it is significantly more uniform and more evenly distributed on the surface than TiAlC + TEAH (Figure 1b). The chemical elements of the TiAlC coating (Ti, Al and C) are mixed, possibly forming various compounds. Both coatings are thin, and the penetration of the coatings' elements reaches 15–20 μm.

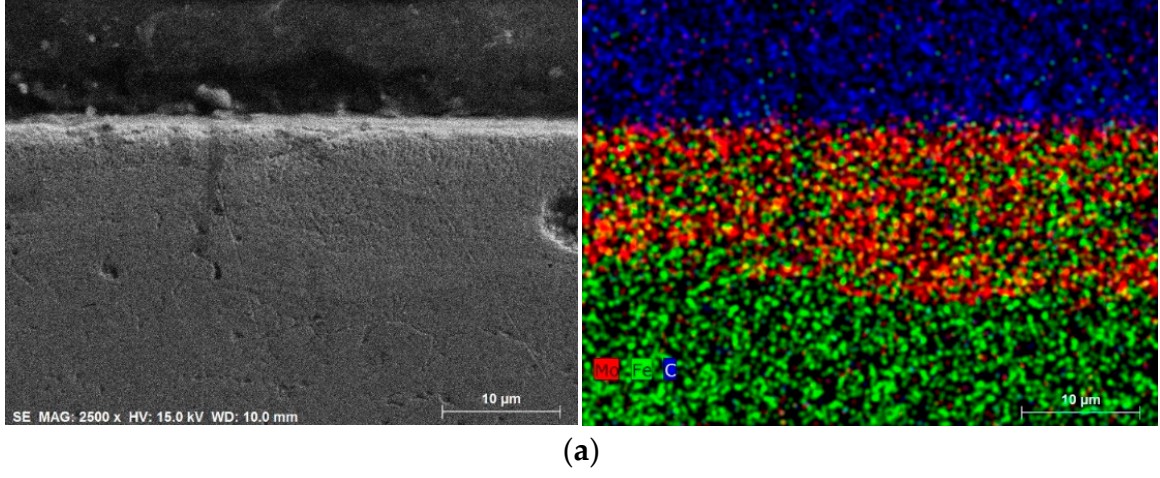

(**a**)

**Figure 1.** *Cont.*

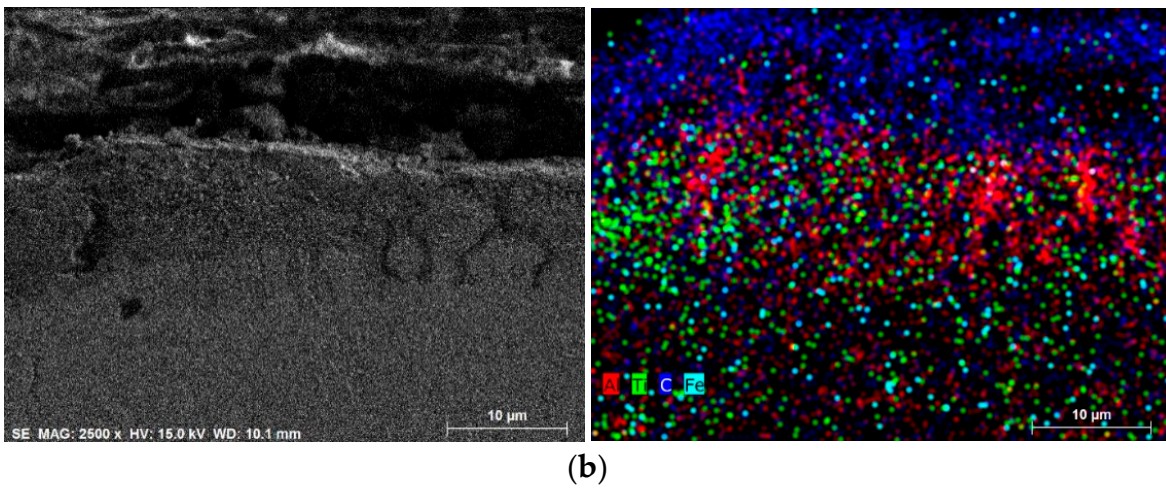

**(b)**

**Figure 1.** SEM images of the general view and distribution of elements in the cross-section of samples: (**a**) Mo + TEAH coating and (**b**) TiAlC + TEAH coating.

### 2.1. Surface Hardness

Because the structure of the Mo and TiAlC coatings was not the same (Mo was more homogeneous), the hardness on the coating surface was measured differently. The Mo coating surface was measured by forming a matrix of $3 \times 3$ points in x, y coordinates with 0.5 mm distance (9 measurements). On the TiAlC surface, the measurement locations were collected visually by measuring several times in each of the visually different structural components. The mean value of the results was taken. As a result of the mentioned structure unevenness, the error of hardness measurement values for the Mo coating was no more than 1%, and for TiAlC—≤3%. The segments with coatings were then cut perpendicular to the surface and measured in the cross-section; five measures were performed along the coating, 10 µm from the surface. Additionally, the instrumented elastic modulus, indentation work ratio ($\eta_{IT}$), and creep ($C_{IT}$) were measured on top of the coatings and recorded. Indentation curves (indentation Pd, µm vs applied indentation load Fn, N) were made when the test was performed on top of the sample and when the test was performed in the section. The microhardness measurement was processed using a micro indentation tester (CSM Instruments-Anton Paar GmbH, Buchs, Switzerland) device at a load of 0.25 N, 30 s of load addition, 10 s pause, and 30 s load removal time.

### 2.2. Surface Roughness

The initial roughness of the coatings depends on discharge energy during the deposition of the electrospark process and the material to be deposited [18,19]. The roughness was measured at the initial coatings, RS, and counter-body (disks) surfaces. The disks for each tribological test (300 N-180 km, 600 N-180 km, test at different loads and rate of rotation speed and RS) were taken from one batch, and roughness parameters Ra was measured because the obtained coatings had a greater roughness than the surface of the machine-turned counter-body (disks) surface. Thus, the coatings and RS were additionally polished to equalize one friction surface to another.

Polishing was executed by hand in two stages. Ultra-fine sandpaper with a grit number of P1500 was used for the first stage, whereas ultra-fine sandpaper with a grit number of P2500 was used for the second stage. The sandpaper was attached to a flat glass table. Each sample was polished for 4–5 min in the first stage and for 1–5 min in the second stage while measuring the resulting roughness. Surface roughness was measured using a stylus profilometer MahrSurf GD 25 (Mahr GmbH, Göttingen, Germany) with a stylus tip of 2 µm radius and a measurement length of 3 mm. The average of five measurements was recorded.

### 2.3. Rinsing and Weighing of the Samples

Before and after the tribological test, each friction pair was rinsed in an ultrasound bath (DT103, Bandelin electronic, Berlin, Germany) for one hour: 30 min in toluene and 30 min in acetone. After rinsing each friction pair, segments and disks were weighed separately using an electronic balance ABJ 120-4M (Kern & Sohn GmbH, Balingen, Germany). The balance readability is 0.1 mg. The segments and disks were weighed six times, and the means of these measurements were recorded.

### 2.4. Tribological Test

As indicated in Figure 2, the tribological test was performed using a modernized friction machine SMC-2 using a "block-on-ring" scheme. Comparative wear tests were performed with lubricating oil SAE 5W-30 that is specific for medium- and high-load friction pairs. The tests were performed in two different stages. Loads of 300 N and 600 N were applied, the friction path length was 180 km, and the constant rotation rate was 600 rpm. The tests for every sample were performed twice, and the mean values were presented. The test was repeated one more time if the error of wear (or friction torque) was more than 5%.

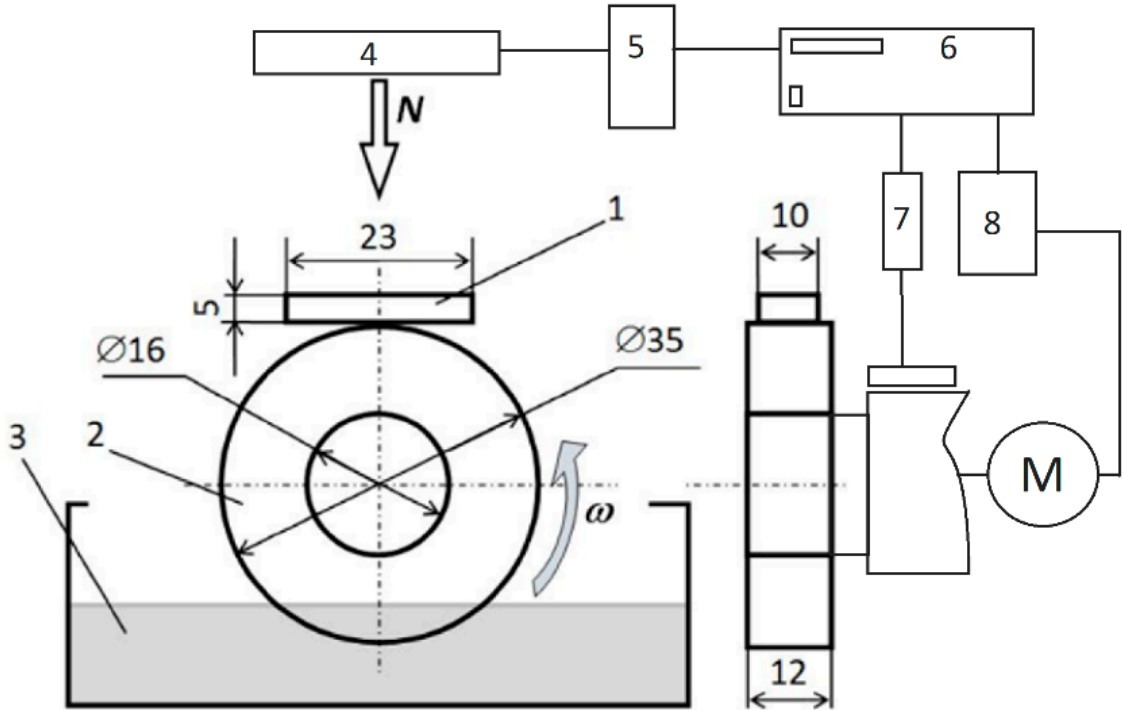

**Figure 2.** Scheme of tribological tests of a friction pair (block-on-roll): 1—block; 2—cylindrical shape steel sample; 3—oil bath; 4—loading system; 5—pneumatic system; 6—computer; 7—torque measurement system; 8—motor drive system; N—normal load; M—motor; $\omega$—roller rotation direction.

The control parameters, such as load, friction torque, rotation speed, and the length of the friction path, were registered during the entire test. The wear of the segments and disks was determined by weight using an electronic balance ABJ 120-4M (KERN & Sohn GmbH, Balingen, Germany) with an accuracy of 0.1 mg. Tribological tests were carried out in the Tribological laboratory of Vytautas Magnus University.

### 2.5. Scanning Electron Microscopy (SEM) and X-ray Diffraction (XRD) Analysis of the Surface

The morphology of the coatings after tribological tests was investigated using scanning electron microscopy (SEM, Hitachi S-3400N). The elemental composition of the samples was obtained using energy-dispersive X-ray spectroscopy.

X-ray diffraction analysis (XRD) was performed on a D8 Advance diffractometer (Bruker AXS, Karlsruhe, Germany) operating at a tube voltage of 40 kV and a tube current of 40 mA. The X-ray beam was filtered with a Ni 0.02 mm filter to select the CuKα wavelength. The diffraction patterns were recorded in a Bragg–Brentano geometry by using a fast counting detector Bruker LynxEye based on the silicon strip technology. The samples were scanned over the range of 2θ = 3–70° at a scanning speed of 0.1 degree/s by using the coupled two theta/theta scan type. Diffrac.eva v4.3 software was used to identify the crystalline phases in the samples.

## 3. Results

The surface characteristics (microhardness, roughness) of the investigated coatings play a determinant role by estimating the factors influencing the tribological characteristics of investigated coatings.

### 3.1. Microhardness and Elasticity Testing

The Mo + TEAH coating is harder than the reference sample, whereas the TiAlC + TEAH coating has a lower hardness than the reference sample. As shown in Figures 3 and 4 and Table 2, the microhardness of the Mo + TEAH coating is higher than the RS and the TiAlC + TEAH coating. Tests on top of the coatings show greater microhardness than those tested in the cross-section. However, for the TiAlC + TEAH coating, the microhardness in the cross-section is higher than on top. ISO14577 defines the parameter indentation work ratio $\eta_{IT}$ [20]. The indentation work ratio is obtained as a ratio of the indentation work by elastic deformation $W_{elast}$ to the mechanical work caused during the entire indentation process $W_{total}$. As can be seen, the $\eta_{IT}$ parameter correlates with microhardness: the lower the microhardness, the lower the elastic index, which indicates the lower the elasticity of the coating [20]. The $C_{IT}$ parameter is higher for both the Mo and the TiAlC TEAH coatings.

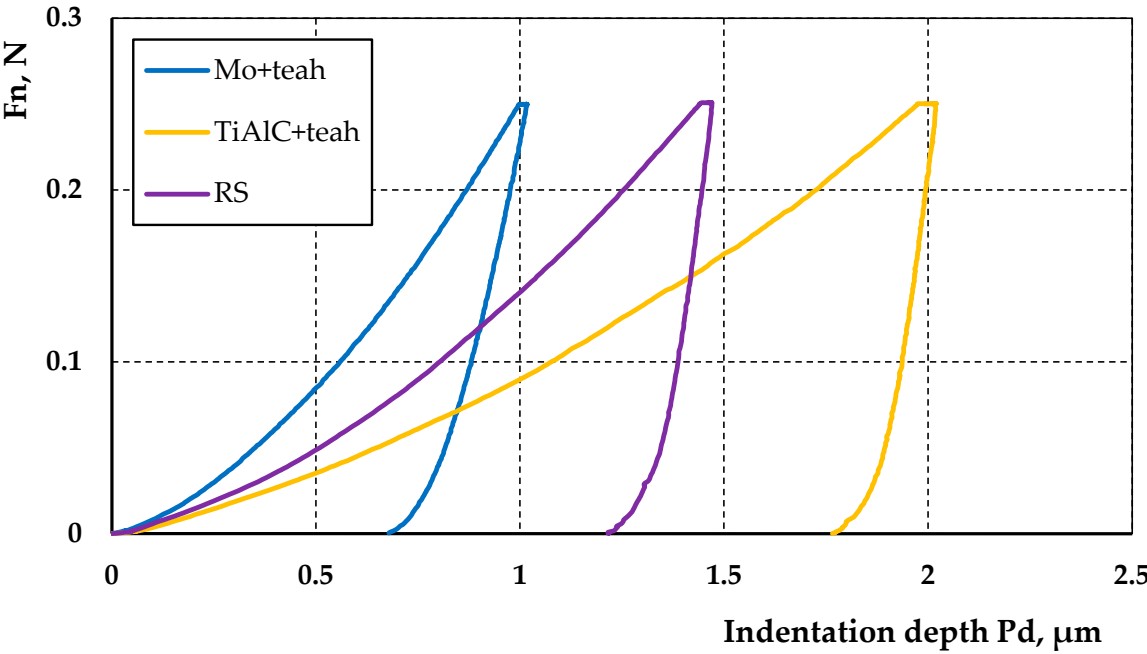

**Figure 3.** Indentation test loading–unloading mean curves measured on top of the coatings and reference sample.

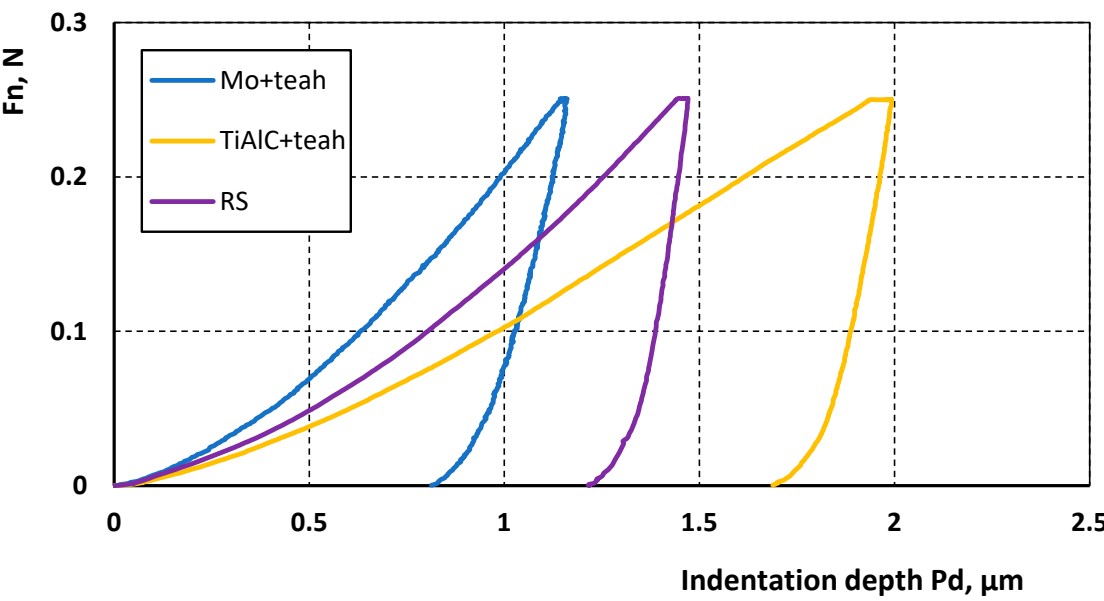

**Figure 4.** Indentation test loading–unloading mean curves of the coatings and reference sample measured in the section.

**Table 2.** Coating and reference sample indentation result at 250 mN.

| Coating | Microhardness, GPa | | Work by Elastic Deformation, $W_{elast}$, µJ | | Total Work of the Process, $W_{total}$, µJ | | Elastic Index $\eta_{IT}$, % | |
|---|---|---|---|---|---|---|---|---|
| | On Top | In Section | On Top | In Section | On Top | In Section | On Top | In Section |
| Mo + TEAH | 11.78 | 8.10 | 0.030 | 0.029 | 0.102 | 0.118 | 29.41 | 24.58 |
| Ti-Al-C + TEAH | 2.47 | 2.99 | 0.020 | 0.028 | 0.217 | 0.227 | 9.22 | 12.33 |
| RS | 5.05 | | 0.019 | | 0.148 | | 12.84 | |

### 3.2. Surface Roughness Results

The Ra parameter of the initial roughness of the ESA coatings and RS was higher than that of the machine-turned disks used as a counter-body (Table 3). It is worth mentioning that additional treatment using the TEAH method has reduced the roughness parameters of the Mo and TiAlC coatings. This trend also remains after the polishing of the specimens.

**Table 3.** Initial surface roughness of the coatings, reference samples, and counter-body and surface roughness after polishing, ready for tests.

| Coatings and Counter-body Surface | The Initial Surface Roughness Ra of the Coatings, µm | The Surface Roughness Ra after Polishing Before the Test, µm |
|---|---|---|
| Molybdenum + TEAH | 1.02 | 0.08 |
| Ti-Al-C + TEAH | 2.13 | 0.14 |
| RS | 0.91 | 0.04 |
| Counter-body disk (steel C45) | 0.12 | |

It is supposed that the lower roughness of the coating or the closer to the counter-bodies roughness, the higher the wear resistance of the coating and, believably, of a complete friction pair, taking into consideration the known dependence of wear resistance on initial roughness [2,13,21].

### 3.3. Tribological Tests Results

The wear of the coatings, the RS, and the disks, as well as the wear of a complete friction pair, was evaluated and presented in Figure 5. During the friction test at 300 N, the obtained

wear results for the friction pairs are as follows: Mo + TEAH < TiAlC + TEAH < RS (Figure 5a). It must be noted that the Mo + TEAH coating did not wear off at all. However, its counter-body disk was worn significantly. TiAlC + TEAH also presented significantly higher wear resistance compared to the reference sample.

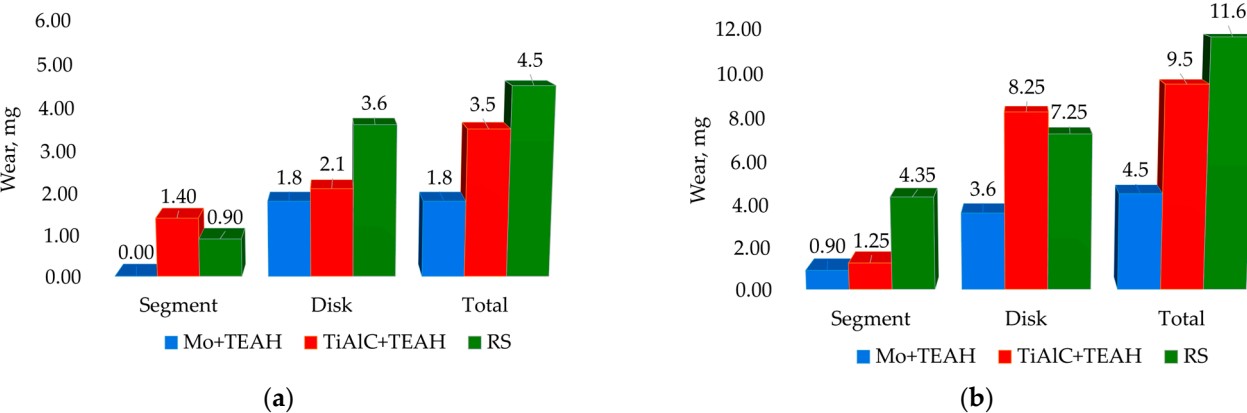

(**a**)　　　　　　　　　　　　　　　　　　　　　　　　　(**b**)

**Figure 5.** Wear results of the tests with the friction path 180 km conducted at (**a**) 300 N and (**b**) 600 N. (**a**)—The error of segment wear did not exceed 1% and the error of disc wear did not exceed 1.5%. (**b**)—The error of segment wear did not exceed 1.5% and the error of disc wear did not exceed 2%.

Figure 5b shows that during the friction test at 600 N, the wear resistance sequence was the same as at a lower loading, but the difference compared to the reference sample here was huge. The TiAlC + TEAH coating had a 3.5 times higher wear resistance, and the Mo + TEAH coating—even 4.8 times more wear resistant. However, in the case of TiAlC + TEAH, the wear of the counter-body was higher than that of the reference sample.

Both coatings show lower total wear values than the reference sample when evaluating the summarized wear values of the specimen and the counter-body. However, using the Mo + TEAH is more efficient—the total value of wear at the tests of this coating is 2.6 times lower than at the tests of the reference sample.

Friction measurements of the coatings (Figure 6) show a similar friction torque value for coatings and reference samples. However, the reference sample shows a trend of increasing friction torque at 300 N loading, and at 600 N loading, it becomes drastically unstable. The Mo + TEAH coating shows comparably higher friction torque at the beginning of the tests, but it has a clear decreasing trend. At the end of the tests, the friction torque equals the other samples' results. The friction values of the TiAlC + TEAH coating are stable for both loading versions.

The summary results of the friction measurements (Table 4) display similar average values and deviations of friction torque for both the coatings and the RS samples at 300 N but slightly higher friction values for the coatings at 600 N. This shows that, regarding the friction losses, the TEAH-processed coatings are more efficient for the friction pairs which operate at lower loading. This could be related to the surface structure formed at such coating, which is favorable for lubrication film formation, but at higher loading, this structure starts to change. When using the oil which is adapted to work under higher load conditions, we assume that frictional losses would be lower. In this study, we did not conduct such an investigation, and it remains for further research.

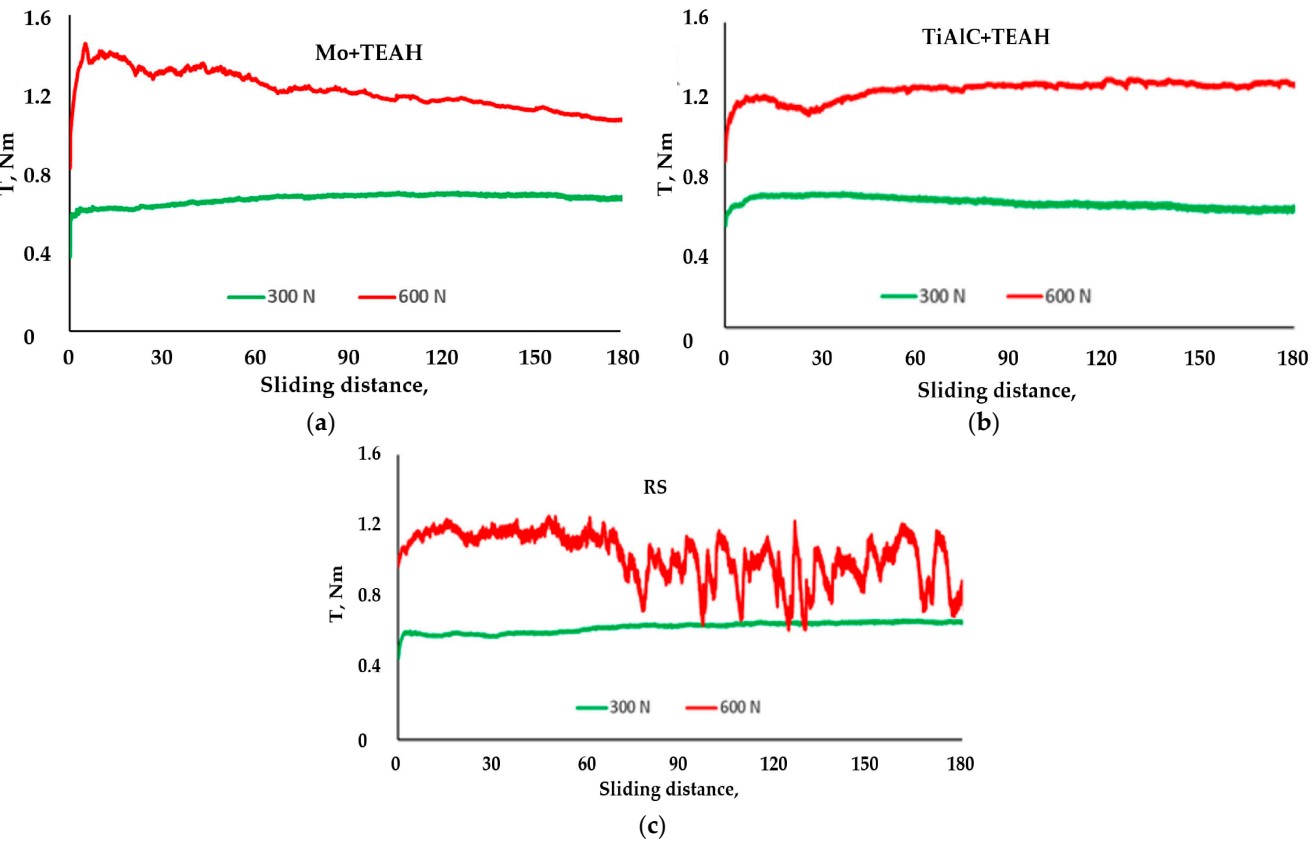

**Figure 6.** The friction torque graphs of each friction pair at 300 and 600 N loads and the rotational speed at 600 rpm. (**a**) MO + TEAH, (**b**) TiAlC + TEAH, (**c**) RS. The friction path was 180 km.

**Table 4.** Summary of friction torque measurements.

| Load, N | Numerical Characteristics | Friction Torque of Tested Samples, Nm | | |
|---|---|---|---|---|
| | | **Mo + TEAH** | **Ti-Al-C + TEAH** | **Not Coated (RS)** |
| 300 N | Average | 0.670 | 0.657 | 0.632 |
| | Max | 0.703 | 0.708 | 0.672 |
| | Min | 0.376 | 0.535 | 0.458 |
| | St DEV | 0.027 | 0.027 | 0.029 |
| 600 N | Average | 1.216 | 1.248 | 1.031 |
| | Max | 1.457 | 1.308 | 1.255 |
| | Min | 0.825 | 0.874 | 0.614 |
| | St DEV | 0.095 | 0.048 | 0.138 |

### 3.4. Surface Analysis of the Coatings by SEM and EDS

The SEM view of the worn paths shows that the predominant type of surface destruction is abrasive. The surfaces are scratched in all cases. The Mo + TEAH coating (Figure 7) indicates a comparably smooth surface with wear traces, some micro-cracks of the coating, and well-seen micro dimples, which are not the result of erupted material but more likely ones formed during the electrospark process. The dimples and micro-cracks could serve as an oil reservoir and improve lubrication between the two surfaces. These circumstances most likely influenced the wear of the disk, which was the smallest among all the samples. The uncoated reference sample surface and the TiAlC + TEAH-coated surface were the most scratched (Figures 8 and 9). The signs of material transfer (adhesion) and fatigue are also characteristic of the RS and the TiAlC + TEAH surfaces. We believe that in the future, it would be appropriate to conduct additional studies using oils containing better EP properties, i.e., able to withstand higher loads.

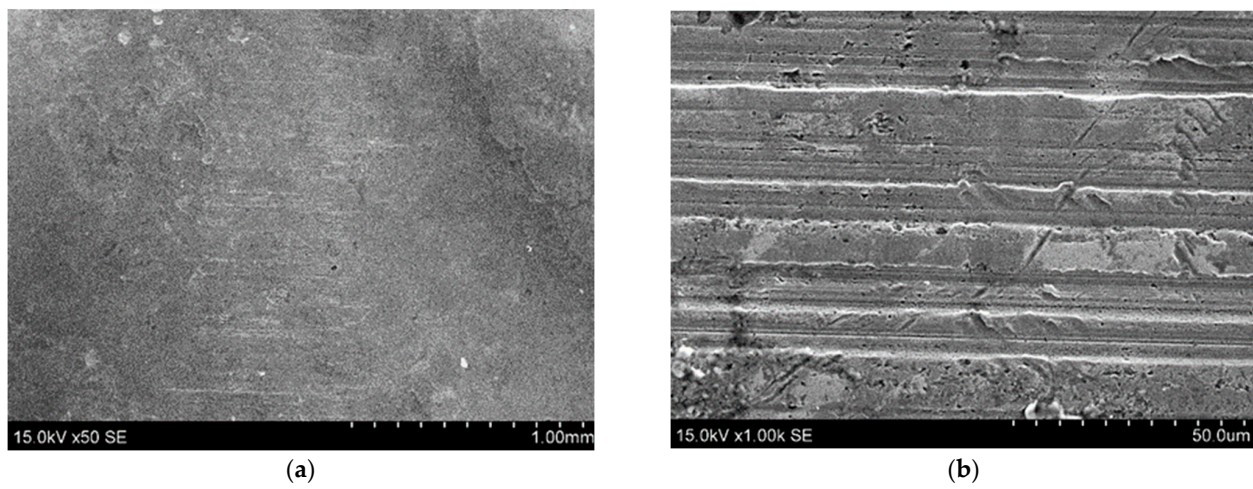

**Figure 7.** SEM images of the coatings Mo+TEAH after a test at 600 N: (**a**) magnification ×50 and (**b**) magnification ×1000.

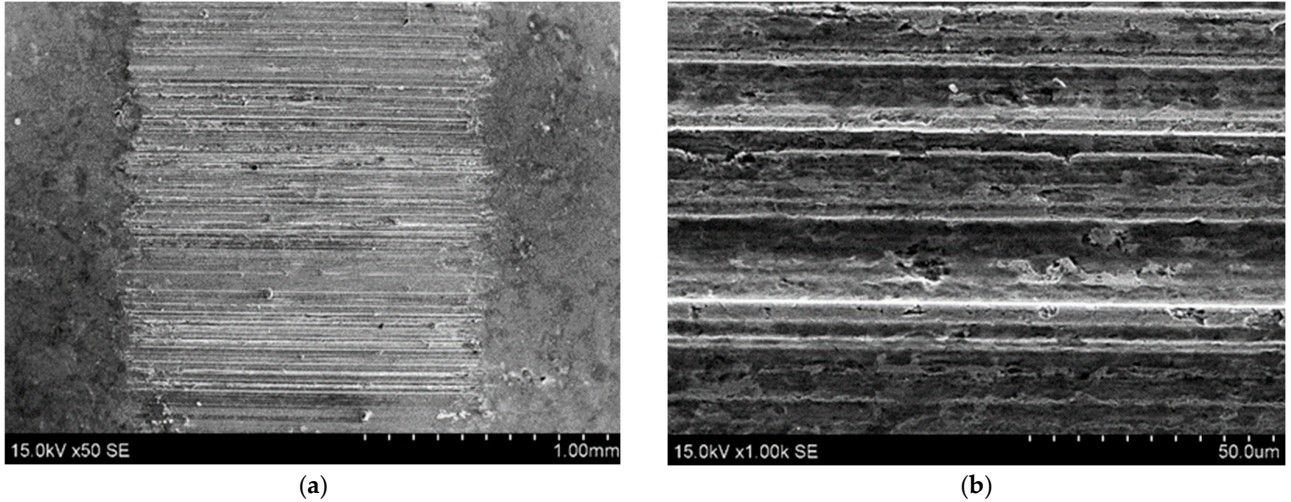

**Figure 8.** SEM images of the coatings TiAlC + TEAH after a test at 600 N: (**a**) magnification ×50 and (**b**) magnification ×1000.

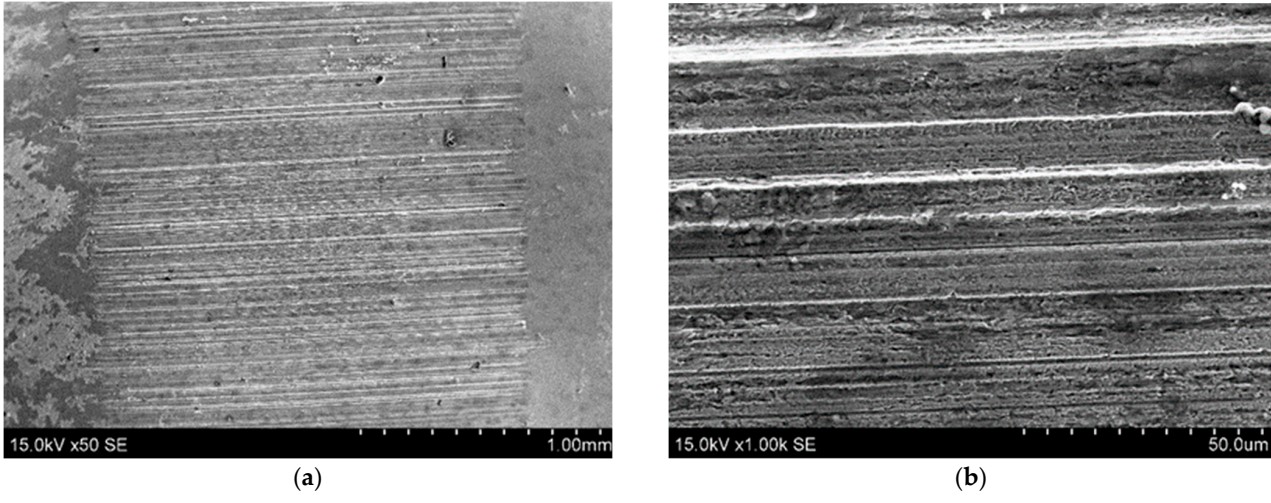

**Figure 9.** SEM images of the uncoated block surfaces after a test at 600 N: (**a**) magnification ×50 and (**b**) magnification ×1000.

EDS images of the tested samples (Figures 10 and 11) display the change of chemical element distribution of the coatings Mo + TEAH and TiAlC + TEAH after the wear tests. The comparison of the Mo + TEAH coating before and after the tests (Figure 10) shows that the distribution of elements is taking place, but the presence of Mo is also clearly seen after the surface is worn. Much more significant elementary changes of the TiAlC + TEAH coated surface take place during this sample's wear (Figure 11). The traces of Al and Ti on the worn surface almost disappear, and carbon and iron are dominant after the tests.

A summary of semi-quantitative EDS analysis was performed according to the surface composition reported for each specimen representing the average of three area scans performed at different magnifications. Because the sampling volume for EDS penetrates through the coating into the substrate material, these data cannot accurately describe the surface coating layer composition. The data demonstrate the presence of a significant amount of Fe, Mo, Zn, Mn, Ti, and Al on all the coated samples, indicating a successful deposition process. In Table 5, we can see the EDS data collected at different surface areas of these coatings.

**Table 5.** Surface composition of reference and coated samples according to EDS analysis.

| Sample | Composition (wt.%) | | | | | |
|---|---|---|---|---|---|---|
| | **Fe** | **Mo** | **Zn** | **Mn** | **Ti** | **Al** |
| C45 | 97.22 | - | - | 0.8 | - | - |
| Mo-TEAH | 71.12 | 19.76 | 2.7 | 0.55 | - | - |
| Ti-Al-C-TEAH | 72.06 | - | 7.23 | 0.76 | 1.74 | 4.4 |

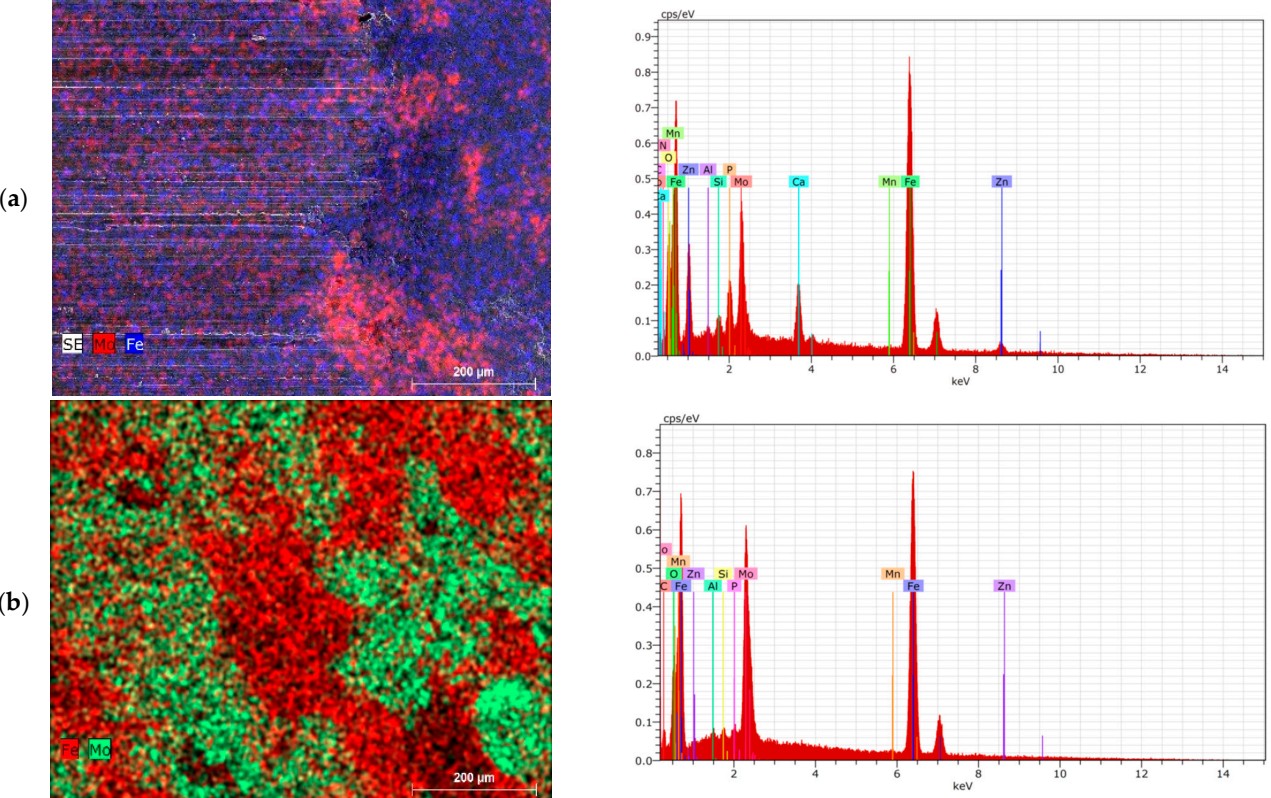

**Figure 10.** SEM images EDS map data and spectra of the wear zone after a test (**a**) and an unworn surface (**b**) of the coating Mo + TEAH.

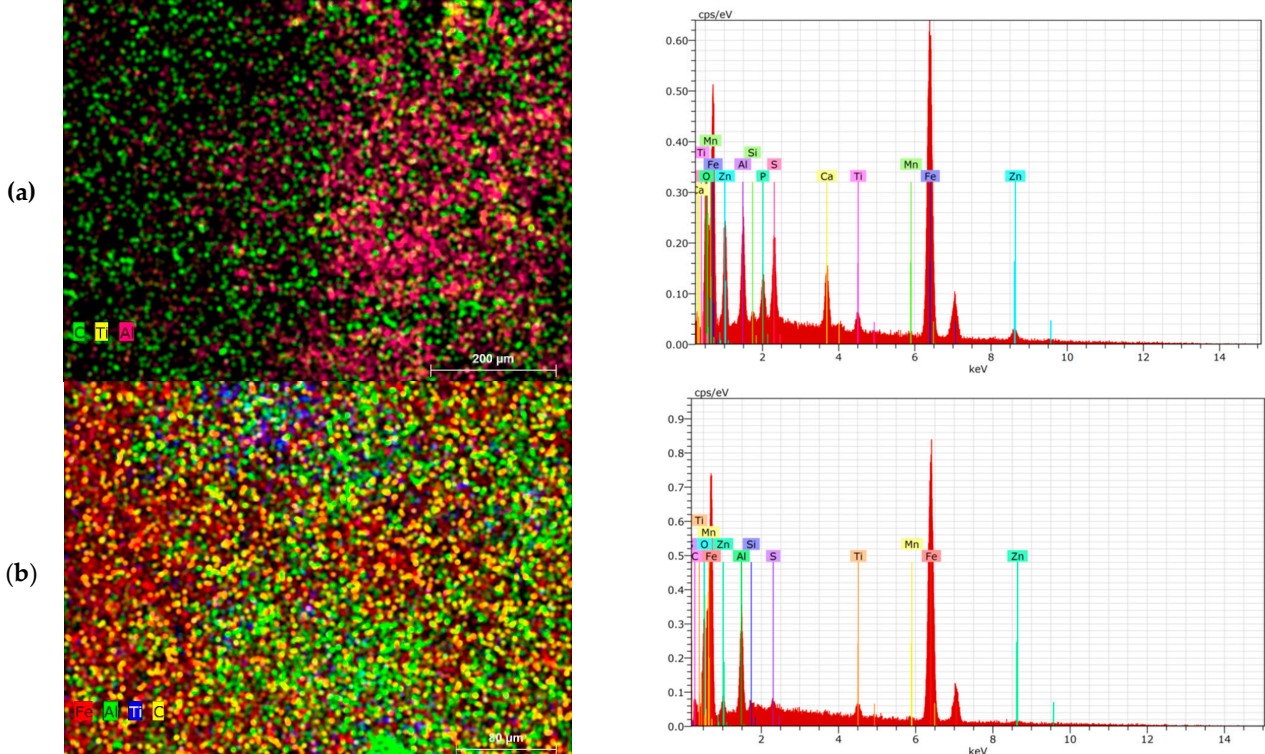

**Figure 11.** SEM images EDS map data and spectra of the wear zone after a test (**a**) and an unworn surface (**b**) of the coating TiAlC + TEAH.

It was determined that the chemical composition (Table 5), structure, and phase composition of the coated samples strongly depended on the methodology and raw materials used in the coating process.

The results of the XRD measurement (Figure 12, curve 1) showed that the steel C45 sample contains only iron (PDF 04-007-9753; d-spacing: 0.202600 nm). It was estimated that in the sample made from steel C45 deposited by the molybdenum + TAEH coating, the broadened peaks typical to iron oxide $Fe_{0.932}O$ (PDF 01-074-1883; d-spacing: 0.215035; 0.248301; 0.152053 nm), zinc iron oxide $Zn_{0.21}Fe_{0.68}O$ (PDF 04-002-0338; d-spacing: 0.215025; 0.248289; 0.152046 nm), aluminum iron $Fe_{0.67}Al_{0.33}$ (PDF 04-003-7166; d-spacing: 0.204283; 0.117943; 0.144450 nm), iron titanium $Ti_{0.103}Fe_{0.897}$ (PDF 04-008-1442; d-spacing: 0.204283; 0.117943; 0.144450 nm), aluminum iron FeAl (PDF 04-002-1309; d-spacing: 0.205768; 0.291000; 0.118800 nm), and iron aluminum carbon $Fe_3AlC_{0.5}$ (PDF 04-005-9528; d-spacing: 0.215929; 0.187000; 0.132229 nm) were identified (Figure 12, curve 2).

It was determined that in the sample made from steel C45 deposited by Ti-Al-C + TAEH coating, iron Fe (PDF 04-007-9753; d-spacing: 0.202600; 0.116971; 0.143260 nm), iron phosphide $Fe_{0.96}P_{0.04}$ (PDF 04-003-5230; d-spacing: 0.202431; 0.116873; 0.143140 nm), iron titanium $Ti_{0.035}Fe_{0.965}$ (PDF 04-004-2489; d-spacing: 0.203435; 0.117453; 0.143850 nm), aluminum iron $Fe_{0.67}Al_{0.33}$ (PDF 04-003-7166; d-spacing: 0.204283; 0.117943; 0.144450 nm), iron zinc $Zn_{0.1}Fe_{0.9}$ (PDF 04-016-7441; d-spacing: 0.204212; 0.117902; 0.144400 nm), iron oxide $Fe_{1.984}O_3$ (PDF 01-077-9925; d-spacing: 0.270122; 0.169566 nm), zinc aluminum iron oxide $ZnFe_{1.7}Al_{0.3}O_4$ (PDF 04-022-6582; d-spacing: 0.252455; 0.296030; 0.148015 nm), iron oxide $Fe_{0.92}O$ (PDF 04-004-7638; d-spacing: 0.214750; 0.247972; 0.151851 nm), and iron oxide $Fe_2O_3$ (PDF 00-013-0534; d-spacing: 0.269000; 0.169000; 0.183800 nm) were formed (Figure 12, curve 3).

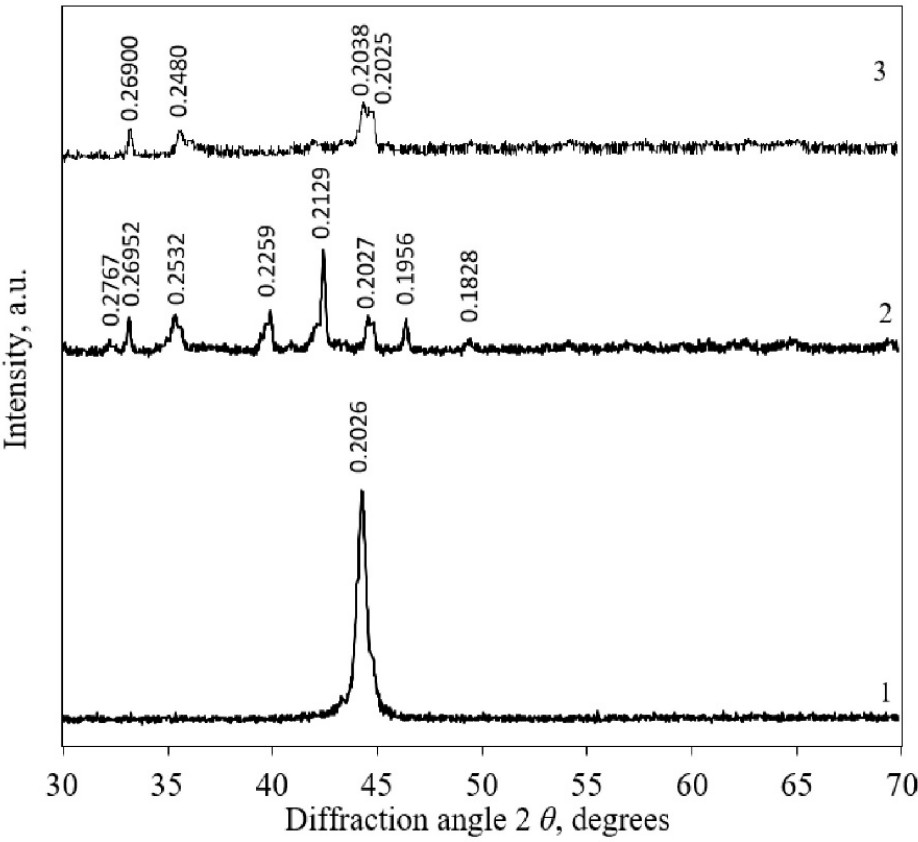

**Figure 12.** XRD spectra of coated samples: 1—C45; 2—Mo + TEAH; 3—Ti-Al-C + TEAH. Indexes: Fe—iron.

## 4. Discussion

The Mo + TEAH friction pair show the best tribological efficiency because of the best wear resistance and the wear of the counter-body disk, which was the lowest among the other samples. In addition to this, the friction torque stabilized at 80 km of the friction path and was stable and low (between 1 N/m and 1.1 N/m) during all tests. Considering the reason for Mo coating wear resistance, it is worth mentioning that its hardness is the highest among other coatings. However, the much higher hardness did not cause increased wear of the softer counter-body—the tested steel disk. This could be explained by the more favorable limitary lubrication conditions of Mo + TEAH compared to the TiAlC + TEAH coating and the reference sample. The friction measurements show that the friction torque reducing trend is even more apparent at the higher loading of 600 N (see Figure 6). Higher loading does not destroy the lubrication advantageous surface structure, and it works even with some abrasion of the surface. The multi-component structure of the TiAlC + TEAH coating shows stable friction results at 300 N loading (it even has some decreasing trends, see Figure 6) but does not have significantly persistent surface structure for good tribological properties at a higher loading, i.e., friction losses start to increase, and wear values upspring significantly, also causing the higher wear of the counter-body.

The surface analysis results confirm these considerations. It shows that the abrasive scratching of all the tested surfaces is taking place. However, in the case of Mo + TEAH, these surface damages are the lowest. This is understandable when considering the highest hardness of these coatings and their ability to retain surface dimples and depressions, which are favorable for lubrication as lubricant reservoirs. Moreover, this property reduces the wear of the counter-body (Figure 5) compared to the other tested specimens.

The EDS and XRD analysis of the chemical elements confirms the stability of this coating, including the presence of Mo, even after wearing the surface at a higher loading.

Using specific high-load adapted lubricant additives which could resist surface damage due to better lubrication abilities, the tribological efficiency of not only the Mo + TEAH but also the TiAlC + TEAH coating could be improved at a higher loading.

## 5. Conclusions

Mo and combined TiAlC coatings with a thickness of 15–20 μm were formed on a steel surface using the electrospark alloying method. Mixing Mo, Ti, Al, and C with the substrate material and each other formed various compounds: iron oxide, zinc iron oxide, aluminum iron, iron phosphide, iron titanium, aluminum iron, etc. The Mo + TEAH and TiAlC + TEAH coatings were formed after processing ESA coatings with thermochemical electrolyte anodic heating. The surface hardness of the Mo + TEAH coating was two times higher than the reference sample, and it was more than four times harder than the TiAlC + TEAH coating.

The friction torque during the tests at a load of 300 N was stable for all variants and for the entire duration of the tests. The loading of 600 N lightly decreased the stability of the friction torque for the Mo + TEAH and TiAlC + TEAH coatings, but testing the reference sample, the higher load caused a drastic reduction of lubrication quality, and there were signs of the surfaces sticking.

The reference friction pair had the highest wear at both loads, and the friction pair with the Mo + TEAH coating wore the least. The total wear, including the counter-body in the testing of the Mo + TEAH coating, was about 2.5 times lower than the control version of the tests.

**Author Contributions:** Conceptualization, J.P. and R.R.; Methodology, M.R.; Formal analysis, A.Ž.; Investigation, M.R., V.M., K.B. and S.T.; Data curation, M.R., K.B. and S.T.; Writing — original draft, M.R., J.P. and R.R.; Visualization, A.Ž.; Supervision, J.P. All authors have read and agreed to the published version of the manuscript.

**Funding:** This research received no external funding.

**Institutional Review Board Statement:** Not applicable.

**Informed Consent Statement:** Not applicable.

**Data Availability Statement:** Data sharing is not applicable to this article.

**Conflicts of Interest:** The authors declare no conflict of interest.

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
