# Peer review of "Investigation of the Lubricated Tribo-System of Modified Electrospark Coatings"

_coatings, doi:10.3390/coatings13050825_

Round 1
Reviewer 1 Report
The paper "Investigation of lubricated Tribo-system of the modified Electro-spark coatings" is suitable for publication in Coatings Journal after some minor corrections. The paper is generally well written, but there are some aspects that need improvement.
1. The abstract is a bit too short. Add an extra paragraph of introduction.
2. In the introduction, the authors should add aspects regarding comparison with other types of coating methods. Also, they should present the influence of chemical elements in terms of microstructure and mechanical properties. Suggested reference:10.1016/j.apsusc.2015.05.111 and 10.3390/mi12121447.
3. SEM and EDS are parameters that need to be improved.
4. The error bars in figure 5 are missing.
5. Add the compounds to the XRD pattern and introduce the ICDD files.
6. Referencing list to short
rest is ok.
Author Response
Dear reviewer,
we sincerely thank you for taking the time to read our article and for your valuable comments and suggestions, that helped us to improve the paper. Below are explanations for your comments.
1. The abstract is a bit too short. Improved. Add an extra paragraph of introduction. Added.
2. In the introduction, the authors should add aspects regarding comparison with other types of coating methods. Also, they should present the influence of chemical elements in terms of microstructure and mechanical properties. Suggested reference:10.1016/j.apsusc.2015.05.111 and 10.3390/mi12121447.
There are shortly mentioned other types of coating methods, like vacuum plasma spraying, laser coating, chemical and physical vapour deposition, arc metallization in the Introduction chapter. The advantages of ESD method are also marked. Reacting to the idea regarding the influence of chemical elements, extra paragraph was added. We describe shortly the influence of Molybdenum to mechanical and tribological properties of the friction surface. The anticorrosion and wear resistance properties of Ti-Al-C coatings are also marked.
3. SEM and EDS are parameters that need to be improved. Improved
4. The error bars in figure 5 are missing. In that scale in which the results are presented the error bars can not be clearly visible, so we present the errors as a table.
5. Add the compounds to the XRD pattern and introduce the ICDD files. Presented
6. Referencing list to short. Improved
Reviewer 2 Report
This is an interesting work but it requires some clarifications as per below
The abstract is Ok despite it was better to present some context
“part of the samples with Molybdenum” not clear that sentence what does means here part of sample ?
It is not clear why you have used C45 against C45 even with one with coating !
Can you give some details why were selected the parameters in Table 1 in such way and not other ?
The polish operation should be described in details for how long it was used each stage. There was any load involved ?
“l SAE 5W-30.” A citation is required for this
Not clear how many repletion for hardness and tribological test were conducted for each type of specimen coated
“measured in section” what is section and top – please make a sketch
Discussions section is interesting but there I found no any citation so how you compare your results with the state of art ?
Some recent references are required
Some more clarity in English style is required.
Author Response
Dear reviewer,
we sincerely thank you for taking the time to read our article and for your valuable comments and suggestions, that helped us improve the paper. Below are explanations for your comments.
1. The abstract is Ok despite it was better to present some context. Added some additional explanation.
2. “part of the samples with Molybdenum” not clear that sentence what does means here part of sample ? The phrase „part of..“ deleted. That was a typing mistake.
3. It is not clear why you have used C45 against C45 even with one with coating ! We chose C45 steel in our research because it is one of the most commonly used steels in mechanical devices and their moving parts. In the tribological research part of our work, we use steel C45 surfaces without coatings as a reference sample, because in this way we want to highlight the positive influence of used coatings on the tribological properties of the friction pair.
4. Can you give some details why were selected the parameters in Table 1 in such way and not other ? Added short explanation.
5. The polish operation should be described in details for how long it was used each stage. There was any load involved ? Described.
6. “l SAE 5W-30.” A citation is required for this. Cited.
7. Not clear how many repletion for hardness and tribological test were conducted for each type of specimen coated. 7.1. “measured in section” what is section and top – please make a sketch. Explained adding some describing sentences.
8. Discussions section is interesting but there I found no any citation so how you compare your results with the state of art? In a specific work, specific electrospark coatings was additionally processed electrochemically. This technology is new, proposed by the authors themselves. We haven't had anything to compare it with yet.
9. Some recent references are required. Comments on the Quality of English Language. Some more clarity in English style is required. Improved.
The manuscript of article we we will submit
Reviewer 3 Report
1)Pls redraw Fig.6 as coefficient vs. sliding distance.
2)Pls change the applied force such as 300N and 600N etc. into pressure.
The English language should be revised by carefully.
Author Response
Dear reviewer,
we sincerely thank you for taking the time to read our article and for your valuable comments and suggestions, that helped us improve the paper. Below are explanations for your comments.
1) Pls redraw Fig.6 as coefficient vs. sliding distance. The friction coefficient offcourse shows the condition of tribocouple. But it is colculated parameter. Foru us was more important the friction torque/force is related to the energetic parameters of tribocouple. So we would not like to change the style of the graphs.
2) Pls change the applied force such as 300N and 600N etc. into pressure. In our case the contact area varies due to wear. So at start position the value of pressure be one and at the end – other. We leave that dimention, moreover, it is often used in articles of this type?
3. Comments on the Quality of English Language. The English language should be revised by carefully. Improved
Round 2
Reviewer 2 Report
Thank you for the revised version.